# Inclusion of initial caries lesions in a population-based sample of Brazilian preschool children: Impact on estimates and treatment needs

**Patrícia de Carvalho**[1]*, **Marcelo Bönecker**[1‡], **Gustavo Tello**[1‡], **Jenny Abanto**[1‡], **Luciana Butini Oliveira**[2‡], **Mariana Minatel Braga**[1]

1 Department of Orthodontics and Pediatric Dentistry, Dental School, University of São Paulo-USP, São Paulo, Brazil, 2 Division of Pediatric Dentistry, Faculdade São Leopoldo Mandic, Campinas, Brazil

☯ These authors contributed equally to this work.
‡ These authors also contributed equally to this work.
* patcarvalho@usp.br

## Abstract

This study aimed to assess changes in epidemiological estimates and treatment needed when initial caries lesions are included in a population-based survey of preschool children. A cross-sectional survey was conducted in a Brazilian municipality, collecting data of pre-school children in 16 health centers. Caries detection used the merged codes (epi-codes) for ICDAS/ICCMS. An option for treatment, according to ICCMS, was chosen during the examination. Caries experience (dmft/dmfs) and prevalence were estimated considering three thresholds (A- initial, moderate, and severe lesions, B- only moderate and severe lesions and C- severe lesions). Incremental need for non-operative care was also verified. The sample consisted of 663 children aged 2–4 years (response rate of 99.85%). Including initial lesions, a 2-fold increase in dmft was observed (A: 3.36, B: 1.02, p<0.001). With the inclusion, the caries prevalence increased to 75% compared to threshold B only (28%). The majority (76%) of children who required any intervention (56%) should be scheduled for non-operative care. We suggest that including initial caries lesions in an epidemiological survey may significantly impact assessment of population caries experience.

## Introduction

National epidemiological surveys to evaluate oral health conditions in children and guide plans in public oral care and public health strategies are essential. In Brazil, this kind of investigation has been carried out since the 1980s. There has been a trend towards a decrease in dental caries indicators, mainly in 12-year-old children, but the same is not evident in the 5-year-old group [1]. A possible explanation could be the absence of prioritization of this age group on the agenda of public health services. Therefore, understanding what may be different in this population is required to guide public health policies.

**Funding:** This study was funded by the Conselho Nacional de Desenvolvimento Científico e Tecnológico (400736/2014-4), Fundação de Amparo à Pesquisa do Estado de São Paulo, and Conselho Nacional de Desenvolvimento Científico e Tecnológico (309817/2015-3, 304319/2018-0). The funders had no role in study design, data collection and analysis, decision to publish, or preparation of the manuscript.

**Competing interests:** The authors have declared that no competing interests exist.

It has been advocated that non-cavitated caries, if detected early, could be arrested through preventive management, so that restorative treatment could be avoided, hence preserving the dental structure and lowering the costs of treatment [2–4]. On the other hand, the national epidemiological surveys do not assess the initial caries lesions. Due to the high number of cavitated caries lesions in Brazil, it is understandable that initial caries lesions are not the priority to guide dental care in public health. Accordingly, the inclusion of these lesions in epidemiological surveys could offer a screenshot of the demand for this type of care and a full understanding of the severity of the disease.

The inclusion of initial caries in epidemiological surveys is possible using ICDAS [5,6]. On the other hand, additional time may be necessary for the examination and less reliable detection of non-cavitated thresholds may be observed [5]. The merged ICDAS codes could be used in epidemiological surveys [7,8]. This methodological approach could simplify the calibration process for epidemiological purposes while guaranteeing the mapping of caries lesions in all severity stages.

Recently, The International Caries Classification and Management System (ICCMS$^{TM}$) was proposed integrating the ICDAS$^{TM}$ as caries process staging to the management of lesions [9]. This system is flexible to merged ICDAS$^{TM}$ scores since some scores require similar management [10,11]. Regarding decision-making, the ICCMS$^{TM}$ advocates the prevention of new caries lesions and controlling the progression of existing lesions. Thus, it preserves the structure of tooth with non-operative care of lesions at early stages and tooth-preserving operative care of severe lesions [9]. Besides, it could guide appropriate public policy for diagnosis.

This study aimed to assess changes in epidemiological estimates and treatment needed when initial caries lesions in a preschool children sample were included. For this purpose, ICDAS merged codes and ICCMS clinical decision-making tree were used. Our study is the first to show this impact on a population-based survey.

## Material and methods

This manuscript was reported accordingly the recommendations of the STROBE Statement [12].

### Ethical considerations

The study has been approved by the Research Ethics Committee, School of Dentistry, University of São Paulo (process number 1.167.931). All parents/caregivers received information regarding the aim of this study and signed informed consent forms.

### Study population and data collection

A descriptive cross-sectional study was undertaken in 2–4-year-old children, living in Maua, São Paulo, Brazil. Maua had a population of 417,064 inhabitants in 2010, including 28,868 children under five years old. The per capita income was approximate US$ 126.24/month, and Human Development Index was 0.766 [13]. The city had a fluoridated water supply (0.7ppm).

As this is the first epidemiological survey in the municipality of Maua, the sample size was calculated based on the prevalence of dental caries in Brazilian 5-year-old children (SB Brazil–National Survey: 53%) [14]. We calculated a sample size to produce a standard error of 5% and a 95% confidence interval. Initially, a sample size of 383 children was estimated. We used the webpage Sampsize to perform calculations. Then, this sample size was corrected by a design effect of 1.4 and increased by an additional 20% to cover non-response. Finally, we considered a minimum sample size of 642 children to be selected.

**Table 1. Merged codes ICDAS[TM].**

| CODE | CLASSIFICATION SEVERITY | CHARACTERISTICS OF LESIONS CONNECTED TO SEVERITY |
|------|------------------------|--------------------------------------------------|
| 0 | Sound | No change in enamel in the plaque accumulation area. |
| A | Initial caries lesion | White spot (translucency other than healthy enamel) or stained fossae (pigmentation) without loss of surface continuity. |
| B | Moderate caries lesion | Cavitation (or loss of surface continuity) located in opaque or pigmented enamel and / or presence of shading of the underlying dentin. |
| C | Severe caries lesion | Cavitation located in opaque or stained enamel with exposure of the underlying dentin. |

Participants were systematically selected from all children attending each of the 16 health centers in 2015 during the National Children's Vaccination Day in Maua, Sao Paulo.

Children were equally selected in all health centers of the municipality. Each fifth child in the vaccination queue was invited to participate. If parents/caregivers did not agree to participate or child showed signs of non-cooperative conduct, the next child in the queue was selected. To avoid possible biases, relatives, and children living in the same household as the selected child were not included in the study. Also, only children whose parents were present were included in the sample to ensure the completion of the questionnaire. Children with systemic and/or neurological diseases and insufficient data should be excluded from the sample. This methodology was used in previous surveys carried out by the same authors [15–19].

Variables referring to socioeconomic conditions such as age, gender, parent's levels of education, and family income were collected.

## Training and calibration

Twenty volunteer examiners (dentists working at Public Service of the municipality) were trained for caries detection using ICDAS[TM] merged codes (Table 1). Examiners were trained and calibrated to evaluate all surfaces of each tooth and classify each one of them according to ICDAS[TM] merged codes and ICCMS[TM] treatment options codes (Table 2). Also, they were trained in clinical decision-making for caries management using ICCMS[TM].

One benchmark examiner conducted the training and calibration sessions, comprising 4 hours each. Sessions included, in the beginning, theoretical explanation and clinical photographic examples. Then, they scored pictures presenting representative scores for both classifications. Subsequently, all the examiners jointly evaluated exfoliated primary teeth set in arch models. They used a dental operating light, 3-in-1 syringe, plane dental mirror, and WHO periodontal probe. The teeth and photos used for the calibrations for the diagnosis had all severity stages of caries. The local Human Bank Teeth donated the used teeth.

This laboratory methodology for calibrating examiners for surveys on dental caries showed to be a feasible alternative to shorten or eliminate the need of examining children several times and it permits to create a wider variety of clinical examples to calibrate the examiners to use the ICDAS [20]. Besides, it seems to provide similar reproducibility figures to those

**Table 2. Clinical caries treatment options based ICCMS[TM] and adapted for the epidemiological study.**

| MERGED CODES -ICDAS | TREATMENT OPTIONS–ICCMS |
|---------------------|-------------------------|
| A | **N:** None (only domicile with fluoride toothpaste > 1000 ppmF). |
| B | **NOC:** Non-Operative Care (professional therapy with fluoride [gel or varnish] and or resin sealants). |
| C | **TPOC:** Tooth Preserving Operative Care (restoration with resin, amalgam or restorative ionomer). |

observers, further, in vivo [20]. This methodology resulted in good results when implemented in the previous surveys [5,6].

Three different combinations of pairs of dental arches were used in the calibration process to simulated children with different caries experience. The arches contained 256 dental surfaces to be evaluated, being 160 (62.5%) sound, 29 (11.3%) presenting initial caries lesions, 12 (4.7%) presenting moderate caries lesions and 55 (21.5%) presenting severe caries lesions, according to ICDAS classification.

## Children's oral examination

The clinical examinations were performed in the dental unit of each health center. Those children who attended to the health center in the Vaccination Day and were selected to participate were examined before the vaccination, to prevent manipulation of the child's oral cavity after receiving the vaccine drops. The clinical examination was conducted on a dental chair using an operating light, a 3-in-1 syringe, plane dental mirror, gauze, WHO ballpoint probe, and individual protection equipment.

A preliminary assessment was performed to assess the presence of urgencies as pulp polyp, ulcer, fistula or abscess (PUFA) [21] and episodes of toothache were reported by parents [22]. Then, teeth were examined as follows.

## Criteria for assessing dental caries

The child's dental surfaces were cleaned with water-soaked gauze, as recommended by Bönecker et al., 2002 [21], and the teeth were evaluated wet and then dried with air from the triple syringe.

The criteria for caries lesions assessment were collected according to the ICDAS$^{TM}$ merged codes for each tooth surface (Table 1) [9]. During the collection, the examiners considered the decayed component according to the ICDAS$^{TM}$ criteria and all dmft components (decayed, filled and missing elements) as proposed by the World Health (WHO) Organization criteria [22].

## Caries treatment needs

There are five key foundation components of the ICCMS™: 1) the staging of the caries process, 2) caries risk classification, 3) the ICCMS™ decision matrices, 4) ICCMS™ comprehensive patient management plan, and 5) Outcomes of caries management using ICCMS™ [9]. Of these keystones, the 1$^{st}$ and the 3$^{rd}$ were considered. Decision-making was done after clinical evaluation of the lesion, assuming that all the lesions were active, since that is the reality for most of the caries lesion in this age group [6]. Caries management options were based on merged scores and followed the decision tree described in Table 2.

In the case of moderate lesions, the decision-making was the most conservative, since evaluations were performed within an epidemiological study, in which there were no radiographic images.

## Statistical analyses

Interexaminer reproducibilities between each trained examiner and the reference examiner were calculated at tooth surface level using a weighted kappa test for ICDAS$^{TM}$ merged codes. The intraclass correlation coefficient (ICC) was calculated with a 95% confidence interval (95%CI) considering the absolute agreement among all trained examiners. Thus, systematic differences among examiners were computed [23].

Caries experience and caries prevalence were estimated in the studied group. The impact of including caries lesions at different severity levels was assessed using three thresholds for all estimates: A- initial, moderate, and severe lesions, B- only moderate and severe lesions, and C- severe caries lesions.

For caries experience, we considered the dmft and dmfs. According to the thresholds, only the component d varied. Then, at each threshold, we counted how many teeth or surfaces had caries to compose the component d. Other components (mf) were fixed for calculations. As caries prevalence, we considered the number of cases (classified in each threshold) by the number of children evaluated. Then, if one child had at least one caries lesions at that threshold, she/he would be considered as a case.

dmft/dmfs mean values were compared among thresholds and age groups using analysis of variance for repeated measurements. Pairwise comparisons of means of dmft and dmfs were also conducted to verify the difference among three thresholds (A, B, and C). The prevalence among thresholds and age groups was compared using the "N-1" Chi-squared test. Bonferroni correction was used for adjustment of significance values in multiple comparisons.

The percentage of children requiring each modality of caries management (none, non-operative care, and tooth-preserving operative care) was also calculated. Distributions in each management modality were compared to others using the chi-squared test. Incremental need for non-operative care was also addressed.

For these analyses, we used the statistical software Stata 13.1 (StataCorp LP, College Station, USA) and MedCalc version 18.9 (MedCalc Software bvba, Ostend, Belgium; http://www.medcalc.org; 2018). The maximum Kappa values possible ($\kappa_{max}$) were calculated given the observed marginal frequencies using the application available in http://vassarstats.net/kappa.html (Lowry, VassarStats: Website for Statistical Computation, <http://vassarstats.net/>. Lowry, Richard. VassarStats: Website for Statistical Computation. http://vassarstats.net/; accessed 11 May 2020).

## Results

After the training on extracted teeth for staging caries, reproducibility values of weighted kappa agreement between trained examiners and reference examiners varied from 0.62 to 0.80, and weighted $\kappa_{max}$ ranged from 0.78 to 0.92. The ICC (absolute agreement) among all examiners was 0.65 (95% CI: 0.61 to 0.70).

Six hundred sixty-four children and parents/caregivers were invited to participate in the study, and a response rate of 99.85% was achieved. Only one child was excluded due to insufficient data collection.

Table 3 shows the socio-demographic characteristics of the sample. The majority of mothers and fathers presented more than 8 years of formal education.

About half of the sample had a family income that corresponded up to two times the Brazilian minimum wage (1 Brazilian minimum wage = US$224.50). Only 4.1% of children presented filled teeth, while 14.2% reported an episode of toothache and 1.2% presented pulp polyp, ulcer, fistula or abscess [21] and 14.2% toothache, characterizing a population with few dental urgencies.

In this sample, the mean dmft (±standard deviation), as defined by the World Health Organization, was 1.08 (3.60) and the respective components–d = 0.94 (3.29); m = 0.05 (0.58); f = 0.09 (0.72). 324 children presented initial caries lesions (49%).

Considering the classification proposed in the study, using ICDAS merged codes, the dmft and dmfs increased considerably from cut-off C (similar to WHO classification) to cut-off A (Table 4)–p<0.001. Including initial lesions, almost a three-fold increase in dmft was observed

**Table 3. Socioeconomic and clinical characteristics of the sample (n = 663).**

| VARIABLES | N (%) |
|---|---|
| **Child's age (years)** | |
| 2 | 219 (33.0) |
| 3 | 225 (34.0) |
| 4 | 219 (33.0) |
| **Child's gender** | |
| Female | 326 (49.2) |
| Male | 337 (50.8) |
| **Mother's education*** | |
| ≤8 years | 135 (20.4) |
| > 8 anos | 498 (75.1) |
| **Father's education*** | |
| ≤ 8 years | 157 (23.7) |
| > 8 years | 404 (60.9) |
| **PUFA** | |
| Absent | 655 (98.8) |
| Present | 8 (1.2) |
| **Pain** | |
| Absent | 569 (85.8) |
| Present | 94 (14.2) |

*n lower than 663 due to missing data

(Table 4). When considering initial caries lesions, mean values would be five times higher than the cases with the C cut-off. The prevalence of caries increased for cut-off A (inclusion of initial lesions) compared to cut-off B and C (Table 4). This trend was observed in all age groups, but the impact of including initial caries lesions presented greater magnitude in younger children, especially 2-year-old children.

Different lowercase letters express statistically significant differences within the estimate among the thresholds. Different Greek letters represent differences among age groups within the threshold for each estimate. Note: different notations were used to symbolize differences among age groups and thresholds separately, but Bonferroni corrections considered both variables s for adjustments in multiple comparisons

**Table 4. dmfs, dmft and caries prevalence by age range.**

| | | dmfs | | | dmft | | | CARIES PREVALENCE | | |
|---|---|---|---|---|---|---|---|---|---|---|
| | | A | B | C | A | B | C | A | B | C |
| AGE | N | Mean (95% CI) | Mean (95% CI) | Mean (95% CI) | Mean (95% CI) | Mean (95% CI) | Mean (95% CI) | % (95% CI) | % (95% CI) | % (95%IC) |
| 2 | 219 | 2.2 aα (1.68–2.75) | 0.54 bα (0.30–0.78) | 0.39 cα (0.15–0.63) | 2.39 aα (1.76–3.01) | 0.44 bα (0.27–0.60) | 0.24 cα (0.11–0.36) | 42.9 aα (0.36–0.49) | 15.52 bα (0.11–0.21) | 8.6 bα (0.05–0.13) |
| 3 | 225 | 3.04 aα (2.64–3.78) | 1.13 bα (0.73–1.54) | 0.77 cα (0.38–1.16) | 3.04 aα (2.55–3.53) | 0.81bα (0.56–1.07) | 0.49 cα (0.26–0.73) | 61.3 aπ (0.54–0.67) | 28.0 bβ (0.22–0.34) | 14.6 cα (0.10–0.20) |
| 4 | 219 | 5.4 aβ (4.38–6.45) | 2.81bβ (2.03–3.59) | 2.10 cβ (1.42–2.77) | 4.66 aβ (3.83–5.49) | 1.81 bβ (1.39–2.22) | 1.20 cβ (0.88–1.53) | 67.5 aπ (0.60–0.73) | 42.0 bπ (0.35–0.48) | 31.0 bβ (0.25–0.37) |
| TOTAL | 663 | 3.54 a (3.17–4.05) | 1.49 b (1.18–1.80) | 1.08 c (0.81–1.36) | 3.36 a (2.97–3.75) | 1.02 b (0.84–1.19) | 0.64 c (0.50–0.79) | 57.3 a (0.53–0.61) | 28.5 b (0.25–0.31) | 18.1c (0.15–0.21) |

A- initial, moderate and extensive stage caries lesions; B- moderate and extensive stage caries lesions; C- extensive-stage caries lesions

When analyzing the decision-making, we observed approximately 40 children required NOC, and in the 2-year age group, more than 50% of the children needed no treatment (N).

Tooth Preserving Operative Care (TPOC) was indicated in a minor part of the cases (3%), and more often in older children (Table 5). The majority (76%) of children in need of any intervention (56%) required non-operative care.

## Discussion

Our main results showed that the inclusion of initial caries lesions and the decision of caries treatment using ICCMS indicated a benefit of NOC for the majority of the 2-to-4-year-old children in need of treatment in this population. These findings emphasize the importance of these systems for caries diagnosing, staging, and management in some populations, instead of methods usually employed in conventional epidemiological surveys.

The introduction of non-cavitated caries lesions improved the sensitivity of caries detection in populations with a low prevalence of cavitated caries lesions since caries lesions may present a low progression rate and are found mostly in initial stages [3]. Indeed, low caries experience was observed in the studied population. dmft indices inferior to the Brazilian figures were observed. This low caries experience could be explained by the parents' socioeconomic status, as family income and level of education were associated with dental caries [24]. Also, socioeconomic and environmental indicators, such as the Human Development Index, measures of access to health services, and availability of fluoridated water supply, could influence dental caries prevalence in specific populations. The municipality of Maua has the Human Development Index of 0.766, which is considered high. It occupies the 131st position in the list of the municipalities of São Paulo and 274th among the Brazilian municipalities [13].

Initial caries lesions, by themselves, exhibit a higher chance either for reversing or progressing compared to sound surfaces [25], and they may be easier arrested compared to cavitated lesions [26]. That is why they should not be underestimated and detected, if possible. The initial lesions alone (without cavitated lesions) were most commonly found in younger children. Probably some of their initial lesions did not have enough time to become cavitated. Caries lesions have a dynamic evolution requiring a long time, usually many months or years. In younger children, initial caries lesions have been considered as a predictor of caries progression [27].

When the initial stage caries lesions were included in the calculation of the estimates, the caries prevalence and caries experience increased, corroborating what had already been described [5–7]. On the other hand, caries detection at this level permitted non-operative treatments for children, especially the younger ones, impacting on demand for treatment. Progression patterns of caries lesions are relevant in decision-making since they guide the treatment choices, especially for the initial lesions where they can be managed by non-operative treatment. Currently, it has been advocated that the possibility of arresting these lesions would

**Table 5. Distribution of number and percentage of treatment decision by age range.**

|  | ICCMS |  |  |  |  |
|---|---|---|---|---|---|
| AGE | N | NOC | NOC + TPOC | TPOC | TOTAL |
| 2 | 126 (57.5) | 77 (35) | 12 (5.5) | 4 (1.9) | 219 (33) |
| 3 | 89 (39.5) | 108 (48) | 22 (9.8) | 6 (2.6) | 225 (34) |
| 4 | 73 (33) | 83 (38) | 53 (24.2) | 10 (4.5) | 219 (33) |
| TOTAL | 288 (43.4) a | 268 (40.4) a | 87 (13.2) b | 20 (3) c | 663 (100) |

N- none; NOC- Non- Operative Care; TPOC- Tooth Preserving Operative Care

lead to fewer cases when operative treatment was necessary and a lower overall cost of services [28].

One relevant problem in including initial lesions in epidemiological surveys, especially in developing countries, it is how to plan actions in the face of the high demands encountered. Usually, a very high prevalence of cavitated lesions is observed, while the initial caries lesions can be detected in almost all children [5,6]. Many Public Health Services are unable to manage this situation successful with limited available human and financial resources.

On the other hand, in such circumstances, the NOC could be extended to more children that do not need TPOC and would be beneficial. Few children in the studied group needed TPOC (operative treatment), and for the majority, the dental treatment was not urgent, as described by PUFA and pain indices. New strategies may be created to guide the public oral health policies for some populations. Not only restorative treatments should be considered, but also approaches that may focus on those children who present only initial caries lesions, for example.

In the study population, for summing up 40% of the children, NOC approaches could be planned in public services. The ICCMS$^{TM}$ guides to preventive strategies and early caries management, such as motivational interview, diet intervention, dental prophylaxis, topical fluoride application, oral hygiene guidance with fluoride dentifrice > 1000ppm, glass ionomer and resin sealants.

Indeed, we must consider caries activity assessment was not performed in this study, because in epidemiological surveys, this criterion exerts little influence on the dental parameters, since most lesions at this age, are active [6,29]. Even though this protocol may lead to the over-treatment of inactive caries lesions, which would probably represent the minority of cases, considering the studied age range. On the other hand, the cost and damage associated with this option would be minimal since the treatment is non-invasive.

The examiners' reproducibility may be another challenge considering the inclusion of non-frankly cavitated lesions in surveys. We observed a substantial to excellent intraexaminer agreement analyzing Kappa and ICC values during the laboratory training. The possible imbalance of marginal totals or not perfect symmetric distribution among them may impact on Kappa values [30,31]. The $\kappa_{max}$ values may be useful information to judge the effect of imbalance in the marginal totals on the magnitude of kappa [30]. Indeed, variations in Kappa (actual and maximum) could be observed, suggesting the presence of such pre-existing factors that could tend to produce unequal marginal totals, e.g. differential sensitivities as using the visual inspect aided by the ICDAS.

The reproducibility values, as Kappa or ICC, should be interpreted with caution since they may be influenced by the prevalence of cases in the sample [30–31]. We consider the prevalence we create in the sample, and the variety of scores included may have influenced these figures. Based on previous observations [3,5], we believe similar, or even higher, values of reproducibility to those observed in laboratory calibration might be found in clinical assessment following such type of training. Clinical and laboratory assessments offered different challenges [3] and one may compensate the other in terms of final results. The prevalence in our sample of extracted teeth was similar to that one observed among children in the survey. Nevertheless, more difficult classifications are often included when training to guarantee examiner's awareness of them. Therefore, even costing appropriate training for that [20], the careful interpretation of reproducibility values under these constraints permits to corroborate the inclusion of initial caries lesions on epidemiological surveys may be a feasible possibility to be used, when necessary.

The present study has limitations inherent to the cross-sectional design, which does not allow establishing a temporal relationship. However, the current findings highlight the

importance of developing public policies in some low-caries progression populations, especially in younger age groups, to identify early lesions and treat them, avoiding their progression and consequent cavitation.

## Conclusion

The inclusion of initial caries lesions may be relevant in an epidemiological survey directed to low-caries progression populations, even in developing countries. It permits a more sensitive map of population needs and, if necessary, the redirection of the public policies in including non-operative care.

## Supporting information

**S1 File. STROBE statement-checklist.**
(PDF)

**S1 Table. ICDAS and ICCMS baseline data.**
(XLSX)

**S2 Table.** dmft in A, B and C threshold.
(XLSX)

**S3 Table.** dmfs in A, B and C threshold.
(XLSX)

**S4 Table. Number of teeth for participants with need of treatment.**
(XLSX)

## Acknowledgments

The authors thank the local authorities (Health Council), City Hall of the Municipality of Maua, the dental examiners, dental nurses, auxiliary nurses, community agents, and the children and their families for their cooperation in carrying out this study. Also, we wish to thank the participants of the Post-Graduate Pediatric Dentistry Seminar of FOUSP for their critical comments.

## Author Contributions

**Conceptualization:** Marcelo Bönecker, Mariana Minatel Braga.

**Data curation:** Patrícia de Carvalho.

**Formal analysis:** Mariana Minatel Braga.

**Investigation:** Gustavo Tello, Jenny Abanto, Luciana Butini Oliveira.

**Methodology:** Marcelo Bönecker, Mariana Minatel Braga.

**Project administration:** Gustavo Tello.

**Writing – original draft:** Patrícia de Carvalho.

**Writing – review & editing:** Patrícia de Carvalho, Marcelo Bönecker, Gustavo Tello, Jenny Abanto, Luciana Butini Oliveira, Mariana Minatel Braga.

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
