## [Decision Letter · Decision Letter 0]

21 Jan 2020

PONE-D-19-32314

Inclusion of initial caries lesions in a population-based sample of Brazilian preschool children: impact on estimates and treatment needs

PLOS ONE

Dear Mrs Carvalho,

Thank you for submitting your manuscript to PLOS ONE. After careful consideration, we feel that it has merit but does not fully meet PLOS ONE’s publication criteria as it currently stands. Therefore, we invite you to submit a revised version of the manuscript that addresses the points raised during the review process.

Dear Dr. Carvalho,

the comments of the reviewer can be found in the attached document. If you have problems to access this document, do not hessitate to contact me.

Thank you fo chosing PLOS ONE

Best wishes

Richard Wierichs

We would appreciate receiving your revised manuscript by Mar 06 2020 11:59PM. To enhance the reproducibility of your results, we recommend that if applicable you deposit your laboratory protocols in protocols.io, where a protocol can be assigned its own identifier (DOI) such that it can be cited independently in the future. For instructions see: http://journals.plos.org/plosone/s/submission-guidelines#loc-laboratory-protocols

We look forward to receiving your revised manuscript.

Kind regards,

Richard Johannes Wierichs, DDS

Academic Editor

PLOS ONE

Journal Requirements:

Reviewers' comments:

Reviewer's Responses to Questions

**Comments to the Author**

1. Is the manuscript technically sound, and do the data support the conclusions?

Reviewer #1: Yes

2. Has the statistical analysis been performed appropriately and rigorously? 

Reviewer #1: Yes

3. Have the authors made all data underlying the findings in their manuscript fully available?

Reviewer #1: Yes

4. Is the manuscript presented in an intelligible fashion and written in standard English?

Reviewer #1: Yes

5. Review Comments to the Author

Reviewer #1: The manuscript is scientifically sound and the methodology well spelt out with minor correction. however, there is need for the author to send to professional english editor for editing.

I have also raised few queries in the manuscript and the authors must address these.

6. PLOS authors have the option to publish the peer review history of their article (what does this mean?). If published, this will include your full peer review and any attached files.

Reviewer #1: Yes: Oyedele Titus Ayodeji

---

## [Author Response · Author response to Decision Letter 0]

6 Mar 2020

Dear Editor in chief,

We are very pleased with the possibility of publishing our research on Plos one.

All reviewers' comments were analyzed and accepted, as in fact they would improve the quality of the article.

In one of the suggestions, the reviewer recommended a review in English by a professional. Therefore, we attached the professional certificate for this review.

We are at your disposal for further clarification.

Sincerely,

The authors

---

## [Decision Letter · Decision Letter 1]

2 Apr 2020

PONE-D-19-32314R1

Inclusion of initial caries lesions in a population-based sample of Brazilian preschool children: impact on estimates and treatment needs

PLOS ONE

Dear Mrs Carvalho,

Thank you for submitting your manuscript to PLOS ONE. After careful consideration, we feel that it has merit but does not fully meet PLOS ONE’s publication criteria as it currently stands. Therefore, we invite you to submit a revised version of the manuscript that addresses the points raised during the review process.

Dear Dr. Carvalho,

Thank you for adressing the raised question. Attached you'll find new reviewer's comments. Although inviting an additional reviewer I think that your manuscript will benefit by addressing the new questions.

Thank you in advance

We would appreciate receiving your revised manuscript by May 17 2020 11:59PM. To enhance the reproducibility of your results, we recommend that if applicable you deposit your laboratory protocols in protocols.io, where a protocol can be assigned its own identifier (DOI) such that it can be cited independently in the future. For instructions see: http://journals.plos.org/plosone/s/submission-guidelines#loc-laboratory-protocols

We look forward to receiving your revised manuscript.

Kind regards,

Richard Johannes Wierichs, DDS

Academic Editor

PLOS ONE

Reviewers' comments:

Reviewer's Responses to Questions

**Comments to the Author**

1. If the authors have adequately addressed your comments raised in a previous round of review and you feel that this manuscript is now acceptable for publication, you may indicate that here to bypass the “Comments to the Author” section, enter your conflict of interest statement in the “Confidential to Editor” section, and submit your "Accept" recommendation.

Reviewer #2: All comments have been addressed

2. Is the manuscript technically sound, and do the data support the conclusions?

Reviewer #2: (No Response)

3. Has the statistical analysis been performed appropriately and rigorously? 

Reviewer #2: Yes

4. Have the authors made all data underlying the findings in their manuscript fully available?

Reviewer #2: (No Response)

5. Is the manuscript presented in an intelligible fashion and written in standard English?

Reviewer #2: Yes

6. Review Comments to the Author

Reviewer #2: I read the paper and I found it very interesting. Overall the manuscript is of interest, however I may rise some comments:

- Materials and methods

Sample size. Taking into account what is written in the introduction and the aim of the paper, there is the possibility that the sample selected is insufficient. The authors should perhaps better explain how they reached the number of 642 children.

Please add dat about no responders, children affected by systemic diseases etc.

The calibration was carried out not clinically, why?

How many examiners? Kappa values have several weak points as described by Feinstein. A.R, Cicchetti. D.V. High agreement but low kappa: I. The Problems of Two Paradoxes. Journal of Clinical Epidemiology 1990; 43: 543-548. Please discuss this issue

The methods are not clear. All the health centers had a dental clinic, too?

How was the examination organized?

Which statistical package was used?

-Results

The absolute agreement recorded was quite low 0.65, please discuss if a possible bias due to this might be present and how to overcome it.

No data regarding the actual disease (dt/ds) of the dmft/s index or about the filling are presented. Moreover, when the authors wrote caries prevalence is caries experience. Please clarify.

7. PLOS authors have the option to publish the peer review history of their article (what does this mean?). If published, this will include your full peer review and any attached files.

Reviewer #2: No

---

## [Author Response · Author response to Decision Letter 1]

16 May 2020

Dear Reviewers, 

We respectfully thank the reviewers for their valuable contributions. Their respective suggestions request for correction that certainly qualify and give greater consistency to our text. 

We describe below the treatment given to each of the amendments proposed by the reviewers for re-submission of manuscript PONE-D-19-32314-R1: "Inclusion of initial caries lesions in a population-based sample of Brazilian preschool children: impact on estimates and treatment needs. Changes in the revised manuscript were highlighted. 

Reviewer #2: I read the paper and I found it very interesting. Overall the manuscript is of interest, however I may rise some comments:

Answer: We are very thankful for the reviewer's feedback on our manuscript. We are going to address each specific comment below. 

- Materials and methods - Sample size. Taking into account what is written in the introduction and the aim of the paper, there is the possibility that the sample selected is insufficient. The authors should perhaps better explain how they reached the number of 642 children.

Answer: To sample calculation, we assumed an expected prevalence of 53%*. This figure was based on data from 5-year-old children in the last National Survey (SB Brasil 2010) since the municipality had not conducted any epidemiologic surveys on caries before. We also assumed precision of 5% to obtain a confidence interval of 95%. Then, a sample size of 383 children was estimated. Finally, this figure was correct by the design effect of 1.4 and, besides, increased by 20% to compensate possible non-responders. Based on these calculations, the minimum sample size required would be 642 children. As children were selected simultaneously in different centers in the National Vaccination Day, a margin of error for each center was considered. In the end, a total of 664 children were selected. Considering the prevalence assumed as expected is close to 50%, we calculated some figure very close to the largest sample size we could achieve (n=385). We used the webpage Sampsize for the calculations. We understood the reviewer's concern about sample size since we used different cut-off points for estimating the prevalence rates. However, during study planning, we considered that and intentionally opted to make the most conservative calculation, based on the previous prevalence of cavitated lesions in Brazilian children, since it can provide the largest sample size possible. Based on the expose, we consider our sample size is adequate to the study purpose. On the other hand, we decided to clarify some aspects about calculations in the manuscript to clarify this methodological step (Page 5) and also included a more informative reference about the national data (reference 14 – Page 20).

(*SB Brasil 2010 – 46.6% caries-free – then, caries prevalence considered as 53.6%.)

Brasil. Ministério da Saúde. Projeto SB Brasil 2010. Pesquisa Nacional de Saúde Bucal. Resultados Principais. Brasília, 2012.

Please add dat about no responders, children affected by systemic diseases etc.

Answer: Actually, we had the response rate on page 10. It was 99.85%. Only one child was excluded due to incompleteness in the filled forms. The reason for non-inclusion was on the same page. No children affected by systemic diseases needed to be excluded, although this was one of our selection criteria at the protocol.

The calibration was carried out not clinically, why?

We used the same methodology from previous surveys in which we obtained good results (1, 2). This methodology showed to be a feasible alternative to shorten or eliminate the need of examinating children several times in the calibration exercises and permits to create a wider variety of clinical examples to calibrate the examiners to use the ICDAS. It seems to provide quite similar reproducibility figures to those observers, further, in vivo (3). Besides, This is specially important considering multiple scores this system requires to be trained, compared with the WHO criteria, for instance. Those are some of the reasons why we chose the laboratory methodology to calibrate the examiners. More detailed advantages may be found in the cited paper (3). A brief justification for the use of this methodology was included in the Methods section (page 5).

How many examiners? Kappa values have several weak points as described by Feinstein. A.R, Cicchetti. D.V. High agreement but low kappa: I. The Problems of Two Paradoxes. Journal of Clinical Epidemiology 1990; 43: 543-548. Please discuss this issue

Answer: Twenty examiners volunteered to participate as examiners in this survey. We apologize for this missing information. It is now included in this version (page 6). We are thankful for the comment about Kappa values interpretation and agree with him. In this version, we described the prevalence of caries in the sample used to calibration exercises and a more detailed description of the calibration process (Page 7). We also calculated (Page 9) and reported the maximum possible Kappa values in the Results section (Page 10), considering possible imbalance of marginal totals or not perfect symmetric among them. Finally, we included a brief discussion about the impact of these occurrences on Kappa magnitude and, possibly, in our calibration results (Pages 18-19) (4, 5).

The methods are not clear. All the health centers had a dental clinic, too? How was the examination organized?

Answer: The examination is described in the item "Children's oral examination", in the Methods section. In the revised version, we clarified that each health center had a dental unit in which the examination was performed (1st paragraph, page 8).

Which statistical package was used?

Answer: For the analyses, we used the statistical software Stata 13.1 (StataCorp LP, College Station, USA) and MedCalc version 18.9 (MedCalc Software bvba, Ostend, Belgium; http://www.medcalc.org; 2018). The maximum Kappa values possible (κmax) were calculated given the observed marginal frequencies using the application available in http://vassarstats.net/kappa.html . We apologize for not having included this information in the 1st version, but we did it in the revised one (Page 10).

-Results

The absolute agreement recorded was quite low 0.65, please discuss if a possible bias due to this might be present and how to overcome it.

Answer: Actually, the absolute agreement among examiners in calibration with extracted teeth may be classified as substantial. This level of examiners' agreement might not be perfect, but acceptable considering the index nature (more than one score to classify caries lesions) and the sample created (to permit examiners to be in contact with a diverse possibility of cases). We consider the prevalence we create in the sample, and the variety of scores included influenced these figures. Even being similar to sample prevalence in the survey, the more difficult classifications are usually added to permit examiner's awareness of them. Based on previous observations, we believe similar, or even higher, figures to those observed in laboratory training might be found in clinical assessment following such type of training. Clinical and laboratory assessments offered different challenges(3), and one may compensate the other in terms of final results. Therefore, even costing appropriate training for that, the careful interpretation of reproducibility values under these constraints permits to corroborate the inclusion of initial caries lesions on epidemiological surveys may be a feasible possibility to be used, when necessary. We included these aspects in the Discussion section, Pages 17-18.

No data regarding the actual disease (dt/ds) of the dmft/s index or about the filling are presented. Moreover, when the authors wrote caries prevalence is caries experience. Please clarify.

Answer: Data about filling may be found on Pages 11-12 when describing the sample according to the presence of fillings, urgent needs and pain episodes. 4.1% of children presented the component f in dmft. In the revised version, we also reported the mean of dmft and its components, using the WHO classification (Page 12). In the present study, we intended to evaluate how using a scoring system to classify lesions with different severities could impact on estimates as caries prevalence and caries experience. As caries prevalence, we considered the number of cases (classified in each threshold) by the number of children evaluated. Then, if one child had at least one caries lesions at that threshold, she/he would be considered as a case. On the other hand, for caries experience, we used the dmft. For the study purpose, according to thresholds, only the component d varied. Then, at each threshold, we counted how many teeth had caries to compose the component d. Other components (mf) were fixed for calculations. We detailed these aspects on Page 9-10 in the revised manuscript.

We are at your disposal for any necessary clarifications. 

Sincerely, 

The Authors 

References mentioned in this letter.

1. Braga MM, Oliveira LB, Bonini GA, Bonecker M, Mendes FM. Feasibility of the International Caries Detection and Assessment System (ICDAS-II) in epidemiological surveys and comparability with standard World Health Organization criteria. Caries research. 2009;43(4):245-9.

2. Braga MM, Mendes FM, Martignon S, Ricketts DN, Ekstrand KR. In vitro Comparison of Nyvad's System and ICDAS-II with Lesion Activity Assessment for Evaluation of Severity and Activity of Occlusal Caries Lesions in Primary Teeth. Caries research. 2009;43(5):405-12.

3. Piovesan C, Moro BL, Lara JS, Ardenghi TM, Guedes RS, Haddad AE, et al. Laboratorial training of examiners for using a visual caries detection system in epidemiological surveys. BMC oral health. 2013;13:49.

4. Sim J, Wright CC. The kappa statistic in reliability studies: use, interpretation, and sample size requirements. Physical therapy. 2005;85(3):257-68.

5. Feinstein AR, Cicchetti DV. High agreement but low kappa: I. The problems of two paradoxes. Journal of clinical epidemiology. 1990;43(6):543-9.

---

## [Editor Report · Decision Letter 2]

20 May 2020

Inclusion of initial caries lesions in a population-based sample of Brazilian preschool children: impact on estimates and treatment needs

PONE-D-19-32314R2

Dear Dr. Carvalho,

We are pleased to inform you that your manuscript has been judged scientifically suitable for publication and will be formally accepted for publication once it complies with all outstanding technical requirements.

With kind regards,

Richard Johannes Wierichs, DDS

Academic Editor

PLOS ONE
---

## [Editor Report · Acceptance letter]

10 Jun 2020

PONE-D-19-32314R2 

Inclusion of initial caries lesions in a population-based sample of Brazilian preschool children: impact on estimates and treatment needs 

Dear Dr. Carvalho:

I'm pleased to inform you that your manuscript has been deemed suitable for publication in PLOS ONE. Congratulations! Your manuscript is now with our production department. 

Kind regards, 

on behalf of

Dr. Richard Johannes Wierichs 

Academic Editor

PLOS ONE